# Air Combat Intention Recognition with Incomplete Information Based on Decision Tree and GRU Network

**DOI:** 10.3390/e25040671

**Published:** 2023-04-17

**Authors:** Jingyang Xia, Mengqi Chen, Weiguo Fang

**Affiliations:** 1School of Management, Wuhan University of Science and Technology, Wuhan 430081, China; 2School of Advanced Interdisciplinary Studies, Hunan University of Technology and Business, Changsha 410205, China; 3School of Economics and Management, Beihang University, Beijing 100191, China

**Keywords:** air combat, GRU network, intention recognition, decision tree, incomplete information

## Abstract

Battlefield information is generally incomplete, uncertain, or deceptive. To realize enemy intention recognition in an uncertain and incomplete air combat information environment, a novel intention recognition method is proposed. After repairing the missing state data of an enemy fighter, the gated recurrent unit (GRU) network, supplemented by the highest frequency method (HFM), is used to predict the future state of enemy fighter. An intention decision tree is constructed to extract the intention classification rules from the incomplete a priori knowledge, where the decision support degree of attributes is introduced to determine the node-splitting sequence according to the information entropy of partitioning (IEP). Subsequently, the enemy fighter intention is recognized based on the established intention decision tree and the predicted state data. Furthermore, a target maneuver tendency function is proposed to screen out the possible deceptive attack intention. The one-to-one air combat simulation shows that the proposed method has advantages in both accuracy and efficiency of state prediction and intention recognition, and is suitable for enemy fighter intention recognition in small air combat situations.

## 1. Introduction

Enemy intention recognition is important in battlefield situation prediction and is a core component of a war decision support system. The prediction of target combat intention is an indispensable link in war and is the basis for understanding battlefield situation and making battlefield decision. As an important means of decision-making, target combat intention prediction or recognition methods have attracted increasing attention from researchers. The recognition of target combat intention refers to the process of recognizing enemy combat intention through reasoning and judgment using relevant methods and comprehensively considering the motion of enemy target, possible combat mission, combat means, and historical combat conditions in a complex and changeable confrontational environment [1]. With the development of informatization technology for weapons and equipment, the battlefield environment has become increasingly complex [2], and battlefield decision-making tends to be intelligent [3]. Compared with ground and sea warfare, it is more difficult to identify enemy intention in air combat owing to its characteristics of strong mobility, wide range of combat, and rapid situational change. The focus of modern war has been shifted from the ground to the air, and the air supremacy determines the outcome of the war to a great extent.

Knowing the enemy intention helps us prepare in advance, which improves the accuracy of decision-making, enhances the operational efficiency of the weapons system, saves war resources, and reduces waste. In this context, many scholars have carried out research on enemy intention recognition and achieved results [4], among which the recognition of air combat intention has become a research focus [5,6]. In particular, with the development of computer and artificial intelligence technologies, air combat intention recognition methods based on intelligent models have been continuously developed and studied [7,8,9,10,11].

In recent years, with the application of an early air warning system, radar stealth composite material and artificial intelligence technology, as well as the uncertainty, incompleteness and immediate change of air combat environment, the research on air combat intention recognition is deepening. In an incomplete information environment, some intelligent systems have difficulty in accurately judging enemy intention. Moreover, due to the complexity and falsity of the battlefield itself, the simple data-driven intelligent models cannot reproduce the battlefield situation and are difficult to make accurate judgments in the complex battlefield environment. This is particularly the case when the enemy deliberately misleads by carrying out false actions; the data-driven intelligent models have obvious obstacle in recognition which can lead to wrong decisions and even falling into enemy traps.

At present, the research on intelligent intention recognition under uncertain and incomplete information in specific air combat scenario is still lacking. The gated recurrent unit (GRU) network was developed based on the improvement of long short-term memory (LSTM) network. Compared with LSTM network, GRU network reduces one gate function while preserving important features through gate control [12]. As a result, GRU network uses fewer parameters. GRU network not only retains the fitting accuracy of LSTM network, but also speeds up the overall training process, which provides it with significant advantages in some scenarios. In recent years, GRU network has been widely used in time series prediction. Because GRU network adopts a simpler architecture than LSTM network, it requires simpler hardware conditions and fewer algorithm components, while greatly improving fitting speed. Shahid et al. [13] used ARIMA, GRU, LSTM, SVR, and other prediction models to predict the time series of confirmed cases, deaths, and recoveries in 10 major countries affected by COVID-19, and compared the advantages and disadvantages of these models. Gao et al. [14] used LSTM, GRU networks, and ANN to simulate the runoff in Fujian Yutan station control catchment from 2000 to 2014. The results show that the prediction accuracy of LSTM and GRU is higher than that of ANN, and the training time of GRU network is the shortest, which has caused GRU network to become the preferred method for short-term runoff prediction. Evidently, battlefield state data are also time-series data. In order to allow the efficiency and accuracy advantages of GRU network in time-series data prediction to reach their full potential, this study adopts a combination of the GRU network and the highest frequency method (HFM) to predict the future state of the enemy fighter in the presence of incomplete air battlefield data. Decision tree is a graphical method that visually uses probabilistic analysis. As a decision support tool, decision tree can effectively assist in formulating optimal strategies [15,16,17]. In addition, as a common tool of machine learning, decision tree has been widely used in all walks of life [18,19,20,21]. To recognize enemy fighter intention, we adopt decision tree to extract intention classification rules from incomplete and uncertain historical data, and then match the predicted enemy state data with the intention classification rules to recognize enemy intention.

The objective of this study is to propose a novel enemy intention recognition method in uncertain and incomplete air combat information environment based on decision tree and GRU network. In particular, we consider the repair of missing data and the detection of deception intention to make up for the shortcomings of existing research. The contributions of this paper include:(1)GRU network and HFM are used to predict the numerical and non-numerical state data of the enemy fighter, and the missing numerical and non-numerical state data are repaired using cubic spline interpolation and mean completer method, respectively.(2)An intention decision tree of enemy fighter is constructed to extract intention classification rules from incomplete and uncertain historical data, where the uncertain data are represented by interval numbers. The index of decision support degree is introduced to judge the node splitting sequence of the decision tree, and the information entropy of partitioning (IEP) is applied to the node splitting criterion. Subsequently, the enemy fighter intention is recognized based on the intention decision tree and the predicted enemy fighter state.(3)The expert experience is integrated into intention recognition, and a target maneuver tendency function is proposed to filter out the deceptive attack intention.

The rest of this paper is organized as follows. Section 2 reviews the literature on battlefield intention recognition. Section 3 describes the problem of air combat intention recognition. Section 4 presents data repair methods for missing state data. Section 5 presents the GRU network model and the modeling process of state prediction. Section 6 constructs the intention decision tree of enemy fighter from incomplete a priori knowledge and presents the intention recognition method based on decision tree. In Section 7, the proposed method is applied and verified in a simulated one-to-one air combat scenario and compared with other methods. Finally, Section 8 summarizes the main conclusions and details future research directions.

## 2. Literature Review

Battlefield situation awareness, situation assessment, and intention recognition are persistent topics in military operations research. The management of data imprecision and uncertainty is becoming increasingly important, especially in battlefield situation awareness and assessment applications, where the reliability of decision-making processes is critical. Rohitha et al. [22] used Dempster–Shafer belief-theoretic relational database (DS-DB) to represent a broader category of data defects, proposed a classification algorithm based on association rule mining, and validated it in a simplified situation assessment scenario. On the other hand, the incompleteness and uncertainty of battlefield situations challenge the efficiency, stability, and reliability of traditional intention recognition methods. The quick and accurate recognition of target tactical intention on the battlefield is a prerequisite for victory in war. Chen et al. [23] proposed a deep learning architecture consisting of a contrastive predictive coding model, a variable length LSTM network model and an attention weight allocator for online intention recognition with incomplete information in war games. They examined the influence of different lengths of intelligence data on recognition performance. As the most common combat mode in modern warfare, the situation assessment, trajectory prediction, intention recognition or behavior prediction of air combat have received extensive attention. Uncertain information exist in every link of air combat situation assessment. Zhou et al. [6] proposed an improved D-S evidence theory framework for the fusion of uncertain information in air combat situation assessment to provide decision-making bases for intention prediction. Bayesian networks have also been used in situation assessment, for example, Xu et al. [5] proposed an improved algorithm for the situation classification of air combat data based on data classification confidence by using semi-supervised naive Bayes classifier.

Target maneuver trajectory prediction is an important prerequisite for air combat situation awareness and threat assessment. Xi et al. [24] proposed a prediction model of target maneuver trajectory based on chaos theory and an improved genetic algorithm-Volterra neural network (IGA-VNN) model, where the chaotic time-series IGA-VNN model was applied to target maneuver trajectory time series prediction. In close-range air combat, highly reliable trajectory prediction can greatly help the pilot win a battle. Zhang et al. [25] proposed an attention-based convolution LSTM memory network to calculate the arrival probability of each space in the reachable region of the target aircraft, which has a higher accuracy than other existing algorithms. Xi et al. [26] developed a target maneuver trajectory prediction model based on phase space reconstruction-radial basis function neural network.

Predicting target intention is helpful to understand the target behavior in advance, thus laying the foundation for air combat decision-making. Zhou et al. [9] proposed an intention prediction method combining LSTM network and decision tree. The future state information of target was predicted from real-time series data based on LSTM network. The decision tree technique was used to extract rules from uncertain and incomplete a priori knowledge. Then, the constructed decision tree was used to obtain the target intention from the predicted data. In addition to LSTM network and decision tree, GRU network and attention mechanism were introduced in air combat intention recognition. Teng et al. [27] and Teng et al. [28] built an air combat intention recognition method based on GRU network, which combined the bidirectional propagation mechanism and the attention mechanism. This method used bidirectional GRU network to carry out the deep learning of air combat features and used the attention mechanism to assign feature weights adaptively. Teng et al. [29] designed a deep learning method attention mechanism based on temporal convolutional network and bidirectional GRU (Attention-TCN-BiGRU) to improve the combat intention recognition of air targets. Comparison with other methods and ablation experiments showed that Attention-TCN-BiGRU outperforms existing methods in terms of recognition accuracy. Addressing the drawbacks of existing air target intention recognition methods, such as timeliness, interpretability, and back-and-forth dependency of intention, Wang et al. [30] proposed STABC-IR method based on bidirectional GRU and conditional random field with space-time attention mechanism. The purpose of intention recognition is to predict the next action or behavior of the opponent. Effectively predicting the behavior of the enemy fighter is crucial to air combat. Yin et al. [31] proposed three patterns to predict fighter behavior. Through the design and implementation of relevant mining/processing algorithms and systems, they found some behavioral experience patterns of fighters and made certain effective predictions of fighter behavior.

Automation and intelligentization have become inevitable trends in modern warfare. Zhang et al. [4] combined the advantages of deep learning and D-S evidence theory to develop an information fusion method for the intention recognition of multi-target formation in sea battlefield. Wang et al. [32] proposed a warship human–machine intelligent interaction model based on the fusion of target intention and operator emotion. Some scholars have applied intelligent model-based air combat intention recognition methods to the battlefield. Xue et al. [8] designed a deep learning method, where a panoramic convolutional LSTM network was proposed in view of the limitation that traditional air target combat intention recognition methods cannot effectively capture the essential characteristics of intelligence information. Ahmed and Mohammed [11] improved the neural network and proposed an attack intention recognition method based on fuzzy min–max neural network.

In recent years, unmanned aerial vehicles (UAVs) or unmanned combat aerial vehicles (UCAVs) are playing an important role in high-tech local wars, where decisions relying on unmanned systems are extremely challenging, as we have seen from the Russia–Ukraine war that broke out in February 2022. Lu et al. [10] proposed an intelligent air combat learning system based on brain learning mechanism, exhibiting greater flexibility in situation assessment and adversary action prediction. When addressing the problem that maneuver trajectory prediction is difficult to maintain high prediction accuracy and short prediction time, Xie et al. [33] proposed a maneuver trajectory prediction method for UAVs based on a layered strategy by combining long-term maneuver unit prediction with short-term maneuver trajectory prediction. Wang et al. [7] presented a robust maneuver decision method with self-adaptive target intention prediction for UAVs, where the reachable set theory and the adaptive adjustment mechanism of target state weights were used to target intention prediction, thus improving the real-time prediction ability. Dong et al. [34] built a motion prediction framework for UAVs, where the target’s future position was inferred according to the current position and estimated direction.

Due to the high complexity of the air battlefield, there is a large space for enemy intention recognition research. However, owing to the uncertainty and incompleteness of information in the actual air combat environment, some intelligent models are not applicable. Moreover, when the enemy deliberately feigns combat moves, the data-driven or model-driven intelligent methods have shortcomings in intention recognition. In the air combat environment with uncertain and incomplete information, intelligent intention recognition is an important issue worth exploring, and false intention recognition cannot be ignored. At present, in the field of air target intention recognition, GRU network is highly valued in target state prediction because of its advantages compared with LSTM network and other methods. At the same time, only a few of the literature related to air combat intention recognition consider missing historical data, and there is currently no existing study on how to detect intentional deception in air combat. In view of this, this study integrates GRU network and decision tree to recognize enemy fighter intention, especially considering the repair of missing data and detection of deception intention, to make up for the shortcomings of existing research.

## 3. Air Combat Intention Recognition Problem

Enemy target intention recognition refers to the process of inferring enemy’s next combat intention by analyzing battlefield information and combining military knowledge and operational basis under dynamic confrontation environment. Enemy target intention recognition is typically accomplished by combining the future state of the enemy target with a priori knowledge.

The current state is represented by real-time data captured by sensors, such as the position, speed, movement direction, radar signal, and maneuver type of the enemy fighter, while the future state is generally obtained by analyzing the current state trend and air combat situation of the enemy. A priori knowledge includes the characteristics and rules of enemy combat state summarized according to historical combat information and empirical laws. The enemy tactical intention usually represents the enemy combat plan and reflects the enemy’s mindset on the battlefield, which cannot be directly observed. However, to achieve a particular tactical purpose, an enemy target must conform to certain laws regarding position, speed, radar signals, maneuver types, and other characteristics. Therefore, by predicting the future state through the collected current state data and then matching the future state with the intention recognition rules based on a priori knowledge, it is expected to recognize the enemy fighter’s intention.

In general, the intention of the enemy fighter in air combat is related to its heading angle, azimuth angle, speed, acceleration, distance, and altitude difference with our fighter, as well as air-to-air radar, air-to-surface radar, electromagnetic interference, and interfered state [9,29,30]. Heading angle, azimuth angle, speed, acceleration, distance, and altitude difference are numerical indicators, that is, the state of these indicators can be represented by specific numerical values. Air-to-air radar, air-to-surface radar, electromagnetic interference and interfered state are all non-numerical indicators and cannot be represented by specific numerical values. In general, these non-numerical indicators have only two typical states: on and off.

Depending on the specific battle type, battle scene, and research objective, there are different ways to describe the battlefield situation graphically. For example, Teng et al. [29] and Wang et al. [30] presented the graphic description of battlefield situation of air-to-ground strike, while Zhou et al. [9] and Teng et al. [27,28] presented the graphic description of one-to-one air combat situation. Considering the combat scene targeted in this study, we present the one-to-one air combat situation as shown in Figure 1, where red represents our side and blue represents the enemy. Unlike other studies, the information transmission and signal interference are also shown in Figure 1 to highlight the information chain under real air combat environment. In Figure 1, *S* is the speed of the enemy fighter. The direction of acceleration is always the same as the speed. *D* is the distance and *Ad* is the altitude difference between the two fighters. The line between the two fighters is the target line of sight, and *A* is the included angle between the target line of sight and due north, known as the target azimuth. *Ha* is the angle between the speed of movement of the enemy fighter and the target line of sight.

To accurately identify the tactical intention of incoming air targets, it is necessary to provide a reasonable tactical intention set of the enemy target. Aerial targets may be fighters, helicopters, UAVs, or missiles. Intention setting often varies greatly depending on ideological background, combat scenario, and target entity. Therefore, the target intention set must be defined according to the corresponding operational context, as well as the basic attributes and possible operational mission of the enemy target. In the context of the enemy air target striking military buildings near shore, Teng et al. [29] established a tactical intention set of enemy air targets, including seven intention types {attack, penetration, surveillance, reconnaissance, feint, retreat, and electronic interference}. Addressing the problem of air target intention recognition, Wang et al. [30] combined the operational context with the attributes and missions of enemy targets to establish a tactical intention set of air targets as {attack, reconnaissance, surveillance, cover, electronic interference, retreat}. Zhou et al. [9] classified air target intention into attack, surveillance, penetration, feint, defense, reconnaissance, cover, and electronic interference for an air combat decision-making system of UCAV. This study is devoted to exploring the intention recognition of an enemy fighter in the operational context of small-scale air combat. Referring to Teng et al. [29], Wang et al. [30], and Zhou et al. [9], based on typical air combat mission types, we establish the tactical intention set of an enemy fighter as {attack, defense, surveillance, penetration, feint, reconnaissance, electromagnetic interference}.

However, in actual air combat, due to the wide application of electromagnetic interference technology and the limitation of sensor transmission capability, as well as the rapid change of operational state itself, it is difficult to completely capture the state data of the enemy fighter, thereby resulting in data loss. To address this problem, we use interval numbers to represent state data that are difficult to obtain accurately, and fix missing data.

In this paper, the intention recognition of the enemy fighter is divided into two parts: state prediction and intention recognition, as illustrated in Figure 2. In the state prediction portion, the state data collected by sensors and other devices are firstly sorted out and the missing data are repaired. The numerical state data are fixed by cubic spline interpolation method, while the non-numerical state data are fixed by mean completer method. Subsequently, GRU network is used to predict the numerical state data, while HFM is used to predict the non-numerical state data. In the intention recognition part, an intention classification decision tree is constructed based on the a priori knowledge of air combat in which the uncertain data are represented by interval numbers. Using the predicted state data as input, the decision tree is retrieved to identify enemy fighter intention. If the intention is identified as attack, the intention is further verified.

## 4. Data Repair

Although information technology is becoming more and more mature, owing to the high complexity of the battlefield, a lot of noise is interspersed in information transmission, resulting in information distortion. At the same time, due to the limitation of the interaction between airborne information and ground sensors, as well as the application of anti-reconnaissance information system, battlefield information acquisition will be intermittently lost, resulting in incomplete information acquisition by sensors. Therefore, to apply the prediction model to the incomplete battlefield information environment, it is necessary to repair the collected data in advance. As mentioned above, the state data of the enemy fighter can be divided into numerical and non-numerical types. The numerical state data mainly refer to the information related to the movement state of enemy fighter. This kind of information is considered time-series data, and the preceding and following data have a certain correlation. The non-numerical state data mainly refer to the state information related to enemy radar with a weak correlation between the preceding and following data. To determine a fast and reasonably accurate data repair method, we compare the applicability conditions of B-spline interpolation [35], Fourier interpolation [36] and cubic spline interpolation [37,38]. According to the characteristics of missing data and the application environment of data repair, we choose the cubic spline interpolation method to fix missing numerical data. For non-numerical data, the correlation is weak due to its small change frequency, so we adopt mean completer method to repair the missing data.

### 4.1. Cubic Spline Interpolation

Cubic spline interpolation is a smooth curve that passes through a series of sample points. Mathematically, the curve-function group can be obtained by solving a three-moment system of equations.

**Definition** **1**[37,38]**.**
*Divide the interval [a,b] into n sub-intervals $\left[(x_{0},x_{1}),(x_{1},x_{2}),\ldots ,(x_{n-1},x_{n})\right]$ with n+1 points, including two endpoints, $x_{0}=a$ and $x_{n}=b$. Let the function values on these points be $f(x_{i})=y_{i}\;(i=0,\;1,\;\ldots ,\;n)$. If S(x) satisfies:*
*(a)* *$S(x)=y_{i}\;(i=0,\;1,\;\ldots ,\;n)$,**(b)* *S(x) is a cubic equation for each subinterval $\left[x_{i},\;x_{x+1}]\;(i=0,\;1,\;\ldots ,\;n-1\right)$,**(c)* *S(x), the first derivative of $\dot{S}(x)$, and the second derivative of $\ddot{S}(x)$ are continuous on the interval [a,b].*
*The function
S(x) can then be constructed as $y=a+bx+cx^{2}+dx^{3}$. S(x) is called the cubic spline interpolation function of f(x) with respect to points $x_{0},\;x_{1},\ldots ,\;x_{n}$.*


### 4.2. Mean Completer

For the repair of non-numerical data, according to the mode principle in statistics, the value with the highest frequency at several moments before and after the missing characteristic value is taken as the repair result. If each state appears at the same frequency at several moments before and after the missing characteristic value, we choose the state that poses the greatest threat to our side as the repair result, so as to avoid underestimating the enemy threat due to incomplete information in air combat, thus strengthening the vigilance against unknown information.

## 5. State Prediction Based on GRU Network

In a fierce battlefield confrontation, a quick response from both sides is crucial. Because GRU network has a similar performance and a faster data processing speed to LSTM network, we choose GRU network to predict the future state of the enemy fighter. GRU network fully retains the advantages of LSTM network in dealing with long-distance dependence, overcoming the problems of gradient explosion and gradient disappearance in training recurrent neural network, and in having a simpler network structure. GRU network integrates three gates of LSTM network into two gates, namely update gate and reset gate, with fewer parameters, thus reducing requirement on training data and improving training speed while ensuring prediction accuracy [13,14]. The typical structure of GRU network is shown in Figure 3 [27,28,29].

In Figure 3, $h_{\left(t-1\right)}$ and $h_{t}$ represent the state information at the previous moment and the current moment, respectively, and $x_{t}$ represents the input information at the current moment. Similar to LSTM network, GRU network can add or remove information through gate control. $z_{t}$ and $r_{t}$ represent the update gate and the reset gate, respectively. Update gate $z_{t}$ is used to control the extent to which the state information at the previous moment is input into the current state. The larger the value $z_{t}$, the more the state information of the previous moment is input. The formula for $z_{t}$ is:(1)$$z_{t}=\sigma_{sig}(W_{z}\times [h_{t-1},x_{t}])$$

Reset gate $r_{t}$ controls the information amount input into the current candidate information ${\tilde{h}}_{t}$ at the previous moment. The smaller the value $r_{t}$, the less information at the previous state is input. The formula for $r_{t}$ is:(2)$$r_{t}=\sigma_{sig}(W_{r}\times [h_{t-1},x_{t}])$$

In Equations (1) and (2), the sigmoid activation function is $\sigma_{sig}(x)=\frac{1}{1+e^{-x}}$; $W_{z}$ and $W_{r}$ are input weight parameters of update and reset gates, respectively. Candidate information ${\tilde{h}}_{t}$ is obtained using the following activation function tanh.
(3)$${\tilde{h}}_{t}=\tanh(W_{\tilde{h}}\times [r_{t}\times h_{t-1},x_{t}]).$$

In Equation (3), $\tanh x=\frac{e^{x}-e^{-x}}{e^{x}+e^{-x}}$, $W_{\tilde{h}}$ is the input weight parameter of candidate information.

Finally, through the update gate, based on the state information at the previous moment $h_{t-1}$ and the candidate information at the current moment ${\tilde{h}}_{t}$, the state information at the current moment $h_{t}$ is obtained as:(4)$$h_{t}=(1-z_{t})\times h_{t-1}+z_{t}\times {\tilde{h}}_{t}$$

It can be seen from Equations (3) and (4) that when both update gate $z_{t}$ and reset gate $r_{t}$ are 1, the information at the previous moment has been completely input into the current information, and the GRU network becomes an ordinary cyclic neural network.

In general, the real-time data of a target in the course of action is a set of time-series data. To overcome the battlefield data deficiency and improve the prediction accuracy, we use a GRU network with *p* input nodes and 1 output node. The first *p* data are used to predict the next data and perform recursive data training. Figure 4 shows the basic framework of the state prediction model.

The framework for predicting enemy fighter state using GRU network consists of the following five steps:(1)Use battlefield satellites, radars, sensors, and other information acquisition equipment to collect time-varying state data of enemy fighter;(2)Repair the missing data in the collected original data;(3)Encode the repaired state data with feature vectors;(4)Input the encoded data into the GRU network for training and obtain the state prediction model;(5)Use the prediction model to predict the state of the enemy fighter at the next moment.

Suppose that the time-series dataset of the target in the course of action is $X=(x_{1},x_{2},\ldots ,x_{n})$ and n>p. In model training, the input of the first training dataset is $X_{1}=(x_{1},x_{2},\ldots ,x_{p})$ and the output is $x_{p+1}$; the input of the second training dataset is $X_{2}=(x_{2},x_{3},\ldots ,x_{p+1})$ and the output is $x_{p+2}$; the last training dataset is $X_{\left(n-p-1\right)}=(x_{\left(n-p-1\right)},x_{\left(n-p\right)},\ldots ,x_{n-1})$ and the output is $x_{n}$. The total training data consist of n−p datasets. After training, the trained model can be used to predict the enemy fighter state at the next moment. The input of the prediction is $X_{\left(n-p\right)}=(x_{\left(n-p\right)},x_{\left(n-p+1\right)},\ldots ,x_{n})$, and the output $x_{n+1}$ is the predicted state at the next moment.

## 6. Intention Recognition Based on Decision Tree

A decision tree is a structure that represents a mapping relationship between object attributes and object values, where each internal node represents a test for an attribute, each branch represents a test output, and each leaf node represents a classification. Decision tree is a kind of supervised learning, which has been widely used in data mining, classification, information retrieval, and prediction. The advantage of decision tree is that it is easy to understand and implement, can handle both data and general attributes, and can produce feasible and satisfactory outputs to large data sources in a relatively short time. The key of the decision tree is the sequence of node splitting and the selection of optimal node splitting criteria. With the rise in decision tree research, fuzzy decision tree [39] and Monte Carlo tree [40] have been proposed successively. Since the calculation of Monte Carlo tree is relatively complicated and time-consuming, and this paper does not involve fuzzy operations, we choose to use the traditional decision tree model.

We use a decision tree to recognize target intention. However, a decision tree containing uncertain or incomplete data is difficult to extract rules from. Therefore, we use interval numbers to represent uncertain data and null values to supplement missing data, thereby matching the uncertainty and incompleteness characteristics of battlefield data. We introduce the index of decision support degree to judge the node splitting sequence in the decision tree and apply the information entropy of partitioning (IEP) to the node splitting criterion.

Suppose that the a priori knowledge of air combat is S=(U, C∪D), where U is the finite non-empty set of statistical objects in the historical data, C is the finite non-empty set of conditional attributes (that is, the state indicators of the target in the historical data), and D is the finite non-empty set of decision attributes (that is, the intention set of the target in the historical data). The a priori knowledge system of air combat is an incomplete information system. We use the symbol * to represent unknown information and use interval numbers to represent uncertain information.

### 6.1. Processing of Incomplete Information

When dealing with incomplete information, this study relies on the concept of similarity relation.

**Definition** **2.**
*Similarity relation [41]: Let S=(U,C∪D) be an incomplete system based on interval-valued attributes, where C is the set of conditional attributes, D is the set of decision attributes, and the symbol * denotes unknown information, which only exists in conditional attributes; that is, *∈C, *∉D. Then, the similarity relation of the conditional attributes SIM(A) (A∈C) defined on U is:*

(5)
$$SIM(A)=\left\{(p,\;q)\in U\times U\left|\forall c\in C,\;f_{c}(p)=f_{c}(q)\;or\;f_{c}(p)=*\;or\;f_{c}(q)=*\right.\right\},$$

*where $f_{c}$ represents the domain of conditional attribute c.*


By the definition of SIM(A), if (p, q)∈U×U is an SIM(A), then p and q are similar, indicating that they have the same property with respect to A.

### 6.2. Decision Support Degree Based on Conditional Attribute

The generation of a decision tree depends on the decision support degree, which depends on the conditional attribute and the optimal split point of attribute interval.

**Definition** **3.**
*Decision support degree based on conditional attribute [9]: Let S=(U, C∪D) be an incomplete system based on interval-valued attributes, U be the finite non-empty set of statistical objects in historical data, C be the finite non-empty set of conditional attributes, and D be the finite non-empty set of decision attributes. *∈C, *∉D, A∈C, $U/A=\left\{A_{1},\;A_{2},\;\ldots ,\;A_{m}\right\}$, $U/D=\left\{D_{1},\;D_{2},\;\ldots ,\;D_{n}\right\}$, $\left|U/A\right|=\sum_{i=1}^{m}\left|A_{i}\right|$. Then, the decision support degree DSD(A,D) of conditional attribute A for decision attribute D is:*

(6)
$$DSD(A,\;D)=1-\frac{\sum_{i=1}^{m}\sum_{j=1}^{n}\left|A_{i}\cap D_{j}\right|\times \left|A_{i}-D_{j}\right|}{\left|U/A\right|\times (\left|U\right|-1)-\sum_{l=1}^{n}\left(\left|D_{l}\right|\times (\left|D_{l}\right|-1)\right)},$$

*where symbol $\left|●\right|$ represents the number of elements in the set.*


Decision support degree DSD(A,D) indicates the support strength of partition U/A to partition U/D. The larger the value of DSD(A,D), the closer U/A is to U/D, indicating that the attribute subset A contributes more to classification, and the greater the certainty of selecting A for classification.

Decision support degree has the following properties:**Property 1:** 0≤DSD(A,D)≤1.**Property 2:** When U/A=U/D, DSD(A,D)=1.**Property 3:** When $U/A=\left\{U\right\}$ and $U/D\neq \left\{U\right\}$, DSD(A,D)=0.

In the air combat scenario, the conditional attribute set is:$$C=\left\{s,\;d,\;\mathrm{Ad},\;\mathrm{Ha},\;\mathrm{Az},\;a,\;\mathrm{Aar},\;\mathrm{Asr},\;\mathrm{Ei},\;\mathrm{Eid}\right\}$$
where s, d, Ad, Ha, Az, a, Aar, Asr, Ei, and Eid are the enemy fighter’s speed, distance, altitude difference, heading angle, azimuth, acceleration, air-to-air radar state, air-to-surface radar state, electromagnetic interference state, and electromagnetic interfered state, respectively.

The decision attribute set, which is the intention set, is:$$D=\left\{\mathrm{Att},\;\mathrm{Def},\;\mathrm{Sur},\;\mathrm{Pen},\;\mathrm{Fei},\;\mathrm{Rec},\;\mathrm{Ele}\right\}$$
where Att, Def, Sur, Pen, Fei, Rec and Ele represent “attack”, “defense”, “surveillance”, “penetration”, “feint”, “reconnaissance”, and “electromagnetic interference”, respectively.

### 6.3. Optimal Split Point of Attribute Interval

**Definition** **4.**
*Split point [9]: If the interval of a finite conditional attribute is $a=[a^{L},a^{U}]$, then the split point $P\;(a^{L}\leq P\leq a^{U})$ is the point that divides the interval conditional attribute into two branches, $a_{1}=[a^{L},\;P]$ and $a_{2}=[P,\;a^{U}]$.*


**Definition** **5.***Information entropy of conditional attribute [42]: Let the decision attribute of conditional attribute A be $D_{A}=\left\{D_{1},D_{2},\;\ldots ,\;D_{k}\right\}$, where k is the number of decision attribute types. Then, the information entropy of conditional attribute A is:*(7)$$I(A)=-\sum_{j=1}^{k}\frac{\left|D_{j}\right|}{\left|D_{A}\right|}\log\frac{\left|D_{j}\right|}{\left|D_{A}\right|}$$*where the symbol $\left|●\right|$ is the number of elements in the set*.

**Definition** **6.**
*Information entropy of partitioning: When the conditional attribute A is divided into $A_{1}$ and $A_{2}$ by split point P, the information entropy of partitioning IEP(A,P) is:*

(8)
$$IEP(A,P)=\frac{\left|A_{1}\right|}{\left|A\right|}\cdot I(A_{1})+\frac{\left|A_{2}\right|}{\left|A\right|}\cdot I(A_{2})$$

*where the symbol $\left|●\right|$ is the number of elements in the set.*


**Definition** **7.**
*Optimal split point. When split point P divides the conditional attribute A and its information entropy of partitioning IEP(A,P) is the smallest among all the split points, then point P is the optimal split point of the conditional attribute A.*


### 6.4. Target Maneuver Tendency Function

On the battlefield, the enemy often makes some false actions consistent with other intentions, deliberately misleading our judgment to achieve other tactical goals. These false actions are highly similar to other intentions in terms of data metrics. Judging only by data metrics, it is easy to fall into enemy trap. Therefore, to further verify the authenticity of enemy fighter intention, we propose a target maneuver tendency function to qualitatively judge the enemy fighter’s action trend. This function comprehensively considers the current battlefield situation, pilot’s operational preference, combat motivation, and kinematics information of the enemy fighter to predict the most likely maneuvering type of the enemy fighter at the next moment. The target maneuver tendency function Pro is defined as:(9)$$Pro=Sit\cdot M\mathrm{ot}\cdot Pre\cdot \frac{s}{s_{\max}}\cdot \frac{a_{\max}}{\left|a\right|+a_{\max}}+K_{\mathrm{Pro}}$$

The greater the value of Pro, the more likely the enemy fighter is to launch an offensive tactical maneuver. The reverse indicates that the enemy fighter is more likely to launch a tactical escape maneuver. When Pro exceeds a certain threshold, the enemy fighter is considered to launch an attack at the next moment. When Pro is less than a certain threshold, it is believed that the enemy fighter is to perform an escape action at the next moment. When Pro is between these two thresholds, it is assumed that the enemy fighter is to maintain tactical motivation. Sit represents the current battlefield situation, which is calculated as:(10)$$Sit=\frac{\sum_{i=1}^{n}E_{ei}}{\sum_{j=1}^{m}E_{fi}}$$

In Equation (10), n represents the number of enemy fighters, m represents the number of our fighters, $E_{ei}$ represents the combat effectiveness of enemy fighter numbered i under the current situation, and $E_{fi}$ represents the combat effectiveness of our fighter numbered j under the current situation.

In Equation (9), Mot is the combat motivation of the enemy fighter represented by a dimensionless number in [0,1]. A large Mot indicates that the enemy fighter is more aggressive, whereas a small Mot indicates that the enemy fighter is more defensive. Pre is the enemy pilot’s operational preference, represented by a dimensionless number in [0,1]. A large Pre indicates that the pilot is aggressive and likely to make offensive moves, while a small Pre indicates that the pilot is cautious and likely to make defensive or escape moves. s represents the current flight speed of the enemy fighter, and $s_{\max}$ represents the maximum available flight speed. $\left|a\right|$ represents the absolute value of the current acceleration, and $a_{\max}$ represents the maximum available acceleration. $K_{\mathrm{Pro}}$ is the correction coefficient of maneuver tendency, which is used to correct systematic errors in the calculation.

### 6.5. Intention Recognition Procedure Based on Decision Tree

The procedure of enemy fighter intention recognition based on decision tree comprises the following seven steps:(1)Determine the interval divisions of conditional attributes by using the a priori knowledge of air combat.(2)Calculate the decision support degree of all conditional attributes, then select the conditional attribute with the highest decision-support degree as the split point.(3)Count all the split points of the conditional attribute, calculate IEP of each split point, and then select the point with the minimum IEP as the optimal split point.(4)Divide the decision information into two parts through the split point, and then divide the other attributes one by one through the above steps until all attributes are divided.(5)Construct decision tree.(6)Based on the predicted state data, the established decision tree is used to judge the enemy fighter intention.(7)If the intention is to attack, the target maneuver tendency function is used to verify the accuracy of the judgment.

## 7. Simulation Study

### 7.1. State Prediction

Since the 1940s, the development of jet fighters is generally thought to have undergone five generations of upgrades. The air combat simulation in this study is aimed at third-generation fighters since they are in service with the largest number of and relatively mature technology. The third generation of fighters entered service in the mid-1960s, represented by F-15, F-16, F-18, Mig-29, Cy-27, and Cy-37 fighters. The third-generation fighters usually adopt a high maneuverability layout and fly at altitudes below 20 km with a maximum flight speed of Mach 2–2.35 (about 650–750 m/s) and cruising speed of Mach 0.9 (about 300 m/s). The maximum available acceleration can reach 70 m/s^2^. However, due to the limitation of the human body’s bearing capacity, the maximum acceleration is generally controlled at 40 m/s^2^. The weapons of the third-generation fighters are mainly medium-range semi-active missiles and combat bombs, and the combat mode is beyond-visual-range attack and close-combat with an effective attack range of up to 120 km.

To demonstrate the effectiveness of the proposed method, we conduct a simulated one-to-one small air combat, in which each side has only one fighter against the other, for example, one F16 fighter against one Cy-27 fighter. We select six numerical indicators, including speed, distance, altitude difference, heading angle, azimuth angle, and acceleration, and four non-numerical indicators, including air-to-air radar state, air-to-surface radar state, electromagnetic interference state, and interfered state, to depict the state of the enemy fighter at different moments. In the simulation, some state information of the enemy fighter is missing, in accordance with the actual battlefield information environment. Assuming that the numerical and non-numerical state data of the enemy fighter at the previous 30 moments have been collected, as shown in Table 1 and Table 2, among which some data are missing. It should be noted that these simulation data are not set arbitrarily, but are based on the aforementioned flight performance and on-board device capabilities of a typical third-generation fighter. The programming language used for calculation and simulation study is Python 3.7.0.

Because the battlefield environment changes quickly, so does data repair, allowing for rapid predictions. We adopt the cubic spline interpolation method to repair the missing numerical state data. Taking the missing data of the speed indicator at moment 5 (node 5) as an example, the speed at node 5 is fitted and repaired according to the speed at the first two nodes and the last two nodes. Through data repair, we obtain the speed at node 5 as 415.4 m/s, as shown in Figure 5, where the smooth fitting curve across the four sample points (nodes 3, 4, 6, and 7) is the cubic spline interpolation curve. It should be noted that the cubic spline interpolation curve is not invariable, but depends on the sample points on which the curve is constructed for particular missing data. Similarly, all missing numerical data are repaired. For the non-numerical data, the mean completer method is used to repair the missing data.

After data restoration, the GRU network is used to predict numerical indicators. Considering the specification of the data volume, we adopt the network structure with three input nodes and one output node. The data at moments 1–24 are taken as the training set, and the data at moments 25–30 as the test set. First, the data are normalized, and then Adam algorithm is used to train the network. This algorithm combines the advantages of momentum algorithm and root mean square prop algorithm, and can adjust the learning rate updating strategy adaptively, thus improving the training speed and accuracy [43]. The mean squared error (MSE) is used to calculate the loss. The number of training iterations is set to 50. The dropout rate, learning rate, and weight decay are set to 0.3, 0.02, and 0.2, respectively. For the non-numerical indicator, the state at the next moment is predicted using HFM; that is, the state with the highest occurrence frequency at the last five moments is taken as the next state.

Once again, taking the speed prediction as an example, Figure 6 shows the training and validation losses of the GRU model, and Figure 7 shows the fitting and prediction results.

As shown in Figure 6, with the increase in training times, both training loss and verification loss decrease rapidly, indicating that GRU network is constantly learning and has good learning effect. After the tenth training, both training loss and verification loss tend to be stable and close to zero, indicating that the model is well fitted on the whole. In Figure 7, because the GRU network has three input nodes and one output node, the fitting curve starts from moment 4, and moment 27 is the predicted moment. As can be seen from Figure 7, the fitting results of the model are in good agreement with the actual speed variation trend. In the training dataset, the maximum fitting error of the model is 5.83%, the minimum fitting error is only 0.01%, and the average fitting error is 2.12%. Table 3 shows the comparison between the predicted speed and the actual speed at moments 25–30 for the test set.

As shown in Table 3, for the test set of the speed indicator, the maximum relative prediction error is 2.61% and the minimum relative prediction error is only 0.02%, indicating that the GRU model has a high fitting accuracy.

To further demonstrate the advantages of GRU network, LSTM network and recurrent neural network (RNN) are applied to the same sample for comparison, and the results are shown in Table 4. The comparative study is performed on the same computer, and the optimal parameters of each model are configured through repeated experiments.

Compared with LSTM network and RNN, GRU network has the advantage in model training time, which is approximately 5 s. In model validation, the root mean square errors (RMSEs) of GRU and LSTM networks are close, while RMSE of RNN is larger. In general, GRU network has better performance in terms of training efficiency and prediction accuracy, and is more suitable for use in the battlefield environment featured with high confrontation and fast response.

In this study, the prediction of each numerical indicator has its own GRU network. Therefore, a total of six GRU networks have been established corresponding to indicators speed, distance, altitude difference, heading angle, azimuth, and acceleration. In fact, each GRU network has only one input parameter and one output parameter of the same type, similar to speed. The prediction performance of GRU network depends on the network training effect. In the case of only one input parameter, the network training effect will depend on the size of the training sample when the network structure is given. Theoretically, there is an optimal sample size for network training. A small sample size that is too small will lead to underfitting and thus reduce the generalization ability of the model, namely the prediction performance, while a sample size that is too large will lead to overfitting and also reduce the prediction performance of the model. In practice, the optimal training sample size of GRU network is closely related to the inherent correlation characteristics of the time series data carried by the network and can be determined through repeated tuning. In this study, according to the training and validation simulation results as well as the comparison analysis, when the training sample size is set to 20–30, the training effect and prediction performance of GRU networks for the six indicators are quite good.

Similar to the prediction of the speed indicator, we use GRU network to complete the training and prediction for the other five numerical indicators. We apply the trained GRU network model to predict the numerical state of the enemy fighter at the next moment beyond the sample. Table 5 shows the predicted state data at moment 31 of the numerical indicators. Table 6 shows the predicted state data at moment 31 of the non-numerical indicators by HFM.

### 7.2. Intention Recognition

We use the historical data to establish the intention decision tree. The 24 sets of simulated historical data containing the state and intention information are given in Table 7 and Table 8.

The numerical state data are then graded according to the historical information. The grading criteria are defined as follows:

Speed: fast [340, 600], medium [200, 340], slow [100, 200];

Distance: long [250, 400], medium [100, 250], short [0, 100];

Altitude difference: high [11, 20], medium [6, 11], low [0, 6];

Acceleration: positive [5, 40], constant [−5, 5], negative [−5, −20];

Heading angle: small [0, 45] and [315, 360], medium [45, 90] and [270, 315], large [90, 270];

Azimuth: north [0, 750] and [5250, 6000], east [750, 2250], south [2250, 3750], west [3750, 5250].

Table 9 lists the grading results.

According to the definition of similarity relation, the incomplete information is processed and the following statistics are obtained.
s_F_ = {4, 5, 6, 9, 10, 12, 13, 14, 15, 16, 17},s_M_ = {1, 2, 3, 4, 7, 8, 11, 14, 21, 23, 24},s_S_ = {4, 14, 18, 19, 20, 22},
where subscripts F, M, and S denote fast, medium, and slow, respectively.
d_L_= {2, 6, 7, 8, 10, 11, 12, 18, 24},d_M_ = {2, 5, 9, 11, 19, 20, 21, 22, 23, 24},d_S_ = {1, 2, 3, 4, 11, 13, 14, 15, 16, 17, 24},
where subscripts L, M, and S denote long, medium, and short, respectively.
Ad_L_ = {1, 2, 3, 4, 5, 9, 13, 14, 15, 17, 20, 23, 24},Ad_M_ = {3, 7, 8, 9, 10, 11, 12, 16, 18, 19, 20, 21, 22},Ad_H_ = {3, 6, 9, 20},
where subscripts L, M, and H denote low, medium, and high, respectively.
Ha_S_ = {1, 2, 3, 4, 5, 7, 8, 10, 11, 16, 17, 19, 22, 23, 24},Ha_M_ = {7, 8, 10, 11, 12, 15, 18, 19, 20, 21},Ha_L_ = {7, 8, 9, 10, 11, 13, 14, 19},
where subscripts S, M, and L denote small, medium, and large, respectively.
Az_E_ = {1, 2, 3, 6, 12, 16, 19, 23, 24},Az_W_ = {2,4,5,6,9,11,12,16,17,21,24},Az_S_ = {2, 6, 7, 10, 12, 14, 15, 16, 20, 22, 24},Az_N_ = {2, 6, 8, 12, 13, 16, 18, 24},
where subscripts E, W, S, and N denote east, west, south, and north, respectively.
a_P_ = {1, 2, 4, 5, 6, 9, 11, 13, 14, 15, 17, 23},a_C_ = {2, 3, 7, 10, 12, 14, 16, 18, 20, 22, 23, 24},a_N_ = {2, 8, 14, 19, 21, 23},
where subscripts P, C, and N denote positive, constant, and negative values, respectively.

Then, according to Equation (6), the decision support degree of each conditional attribute to the decision attribute is calculated. We have:DSD(s, I) = 0.6310,DSD(d, I) = 0.6389,DSD(Ad, I) = 0.5686,DSD(Ha, I) = 0.5961,DSD(Az, I) = 0.6182,DSD(a, I) = 0.5846,DSD(Aar, I) = 0.3521,DSD(Asr, I) = 0.3503,DSD(Ei, I) = 0.4470,DSD(Eid, I) = 0.4719.

Evidently, the distance indicator has the highest decision support degree. Therefore, we partition the decision tree starting from “distance”. Firstly, the split points of distance indicator are counted, and then IEP values are calculated using Equation (8). As shown in Table 7, by aggregating the endpoints of all distance intervals with the repeated endpoints removed, we obtain the following ascending sequence: 40, 50, 60, 70, 80, 90, 100, 110, 120, 130, 150, 180, 200, 210, 220, 255, 270, 280, 290, 300, 310, 330, 350. Then, the optional split points include 45, 55, 65, 75, 85, 95, 105, 115, 125, 140, 165, 190, 205, 215, 237.5, 262.5, 275, 285, 295, 305, 320, 340. Table 10 lists the IEP values corresponding to the split points.

The split point with the minimum IEP is 115. Therefore, it is the optimal split point under the distance indicator. The historical information is then partitioned with split point 115. This procedure is repeated until the final decision tree is generated, as shown in Figure 8. Based on the existing historical statistical information, the established intention decision tree of the enemy fighter consists of 29 nodes, consisting of 1 root node and 14 leaf nodes associated with intention. Each node is represented by a multivariate array, with each number representing a set of historical data. The number in each node circle in Figure 8 represents the serial number of the node, where node 1 represents the root node. The decision tree covers all 24 known historical datasets.

Then, the state data predicted in Table 5 and Table 6 are input into the decision tree to judge the enemy fighter intention. As can be seen from Table 5, the distance to the enemy fighter is 92.31 km, hence the intention judgement goes to node 2. Since the speed is 266.87 m/s, the intention judgment goes to node 5. Again, the heading angle is 1.43° and the intention judgment progresses to node 11. Finally, the azimuth is 2216.15 mil, and the intention judgment terminates at leaf node 20. At this point, the intention to attack has been recognized.

### 7.3. Intention Verification

When the enemy fighter intention is recognized as attack, the recognition accuracy needs to be further tested. In this simulation study, we set the target maneuver tendency threshold as 0.5. When Pro>0.5, the enemy fighter tends to attack; the higher the Pro value, the more obvious the enemy tendency to attack. When Pro<0.5, the enemy fighter tends to escape; the smaller the Pro value, the more obvious the enemy tendency to escape.

Assuming that the enemy pilot’s operational preference Pre and the combat motivation of the enemy fighter Mot are unknown, their values are then set to 0.5. Through expert estimation, the current battlefield situation Sit = 0.35. The maximum available flight speed of the enemy fighter $s_{\max}$ = 600 m/s, the maximum available acceleration $a_{\max}$ = 40 m/s^2^, and the correction coefficient of maneuver tendency $K_{Pro}$ = 0.375. In addition, according to the prediction in Table 5, at moment 31, the current flight speed of the enemy fighter s = 266.87 m/s, and the absolute value of the current acceleration $\left|a\right|$ = 5.83 m/s^2^. Substituting the above parameters into Equation (9), we obtain the target maneuver tendency function Pro = 0.41. This indicates that the enemy fighter has no obvious tendency to attack. Eventually, we correct the enemy fighter intention from attack to feint.

As can be seen from the above results, the enemy fighter intention is identified as “attack” without intention verification. After verification through the target maneuver tendency function, the enemy fighter intention is corrected to “feint”. These two kinds of intentional actions are so similar that it is difficult to distinguish them using only traditional data-driven prediction models, especially when training data are limited and model learning is insufficient. Evidently, these two kinds of intention recognition would result in the subsequent battlefield decisions being completely different. The proposed method of intention recognition in uncertain information environment can effectively predict enemy fighter intention and correct possible misjudgment through secondary recognition, thus further improving the practicality and accuracy of intention recognition, which is critical to correct real-time battlefield decision-making.

## 8. Conclusions

Recognizing the enemy intention is an important prerequisite for making correct and timely battlefield decisions. However, because battlefield information is often incomplete, uncertain, or deceptive, it is difficult to accurately recognize enemy intentions. Especially in air combat, where the information environment is highly complex and rapidly changing, recognizing enemy intentions is even more challenging.

This paper explores an intelligent recognition method of enemy fighter intention in small air combat under uncertain and incomplete information environment. In the presence of incomplete information, GRU network supplemented by HFM is used to predict the future state of an enemy fighter. A decision tree of enemy fighter intention is constructed to extract the intention classification rules from incomplete a priori knowledge. The node splitting sequence of decision tree is determined according to the decision support degree of attributes following the criteria of IEP. Then, the established intention decision tree and predicted state data are exploited to recognize the enemy fighter intention. In particular, to identify the possible deceptive attack, a target maneuver tendency function is proposed to rejudge the attack intention, thus improving the accuracy of intention recognition. In addition, we propose practical data repair methods to solve the unavoidable data missing problem in the air combat information environment. The simulation study shows that the proposed method is suitable for uncertain and incomplete air battlefield information environment, and can screen false attack intention. This method has advantages in both the accuracy and efficiency of state prediction and intention recognition, resulting in potential application value for intention recognition in small air combat situations.

In actual air combat, enemy state prediction, enemy intention recognition, our response action, and enemy state change are mutually influenced, tightly coupled, and alternate. For the specific application of the proposed method, the method can be programmed and embedded in ground or airborne equipment as a module of C^3^I (command, control, communication, and intelligence) system. The intention decision tree is relatively fixed, which can be constructed according to a priori knowledge of air combat summarized from actual air combat or exercise, and updated regularly with the accumulation of a priori knowledge. State data acquisition is enabled when the enemy fighter is detected. After a certain amount of state data is accumulated, the state prediction model can be built to predict the future state of the enemy fighter. Different from the decision tree used for intention recognition, the state prediction model needs constant iterations and real-time updates. For example, the state prediction model can be set as to always be trained by the state data of the latest *n* moments. To this end, the proposed methods must also be supported by advanced information and communication technologies if they are to be practical in actual air war.

Generally, to reduce the burden of data collection, the selected enemy fighter state indicators should be independent from each other as far as possible, and their own state prediction models should be established, respectively. One advantage of the proposed intention recognition framework brought about by modularity is that it is not necessary to make extensive adjustments to the entire prediction model system due to indicator increase or decrease, but only to make an addition or deletion to the independent state prediction model.

Our study can be extended from the following aspects. First, at present, we use relatively simple interval numbers to represent uncertain state data when constructing the intention decision tree. In the future, we will introduce a fuzzy set theory or probability theory to represent uncertain state data, thus constructing a fuzzy or probabilistic intention decision tree. Second, modern war has evolved into an all-round systematic confrontation. Our study only focuses on the most basic antagonistic unit in air combat, namely the one-to-one air combat scenario. In the future, we will explore enemy fighter intention recognition in large air combat under systematic confrontation.

## Figures and Tables

**Figure 1 entropy-25-00671-f001:**
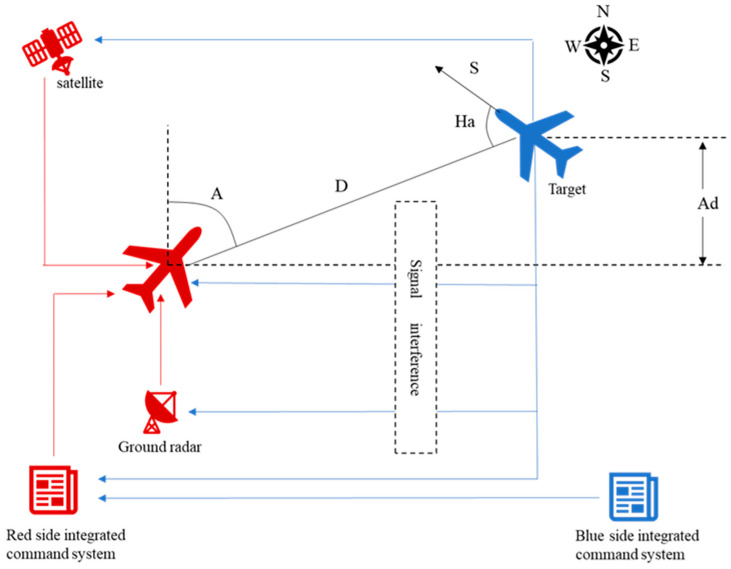
Air combat situation.

**Figure 2 entropy-25-00671-f002:**
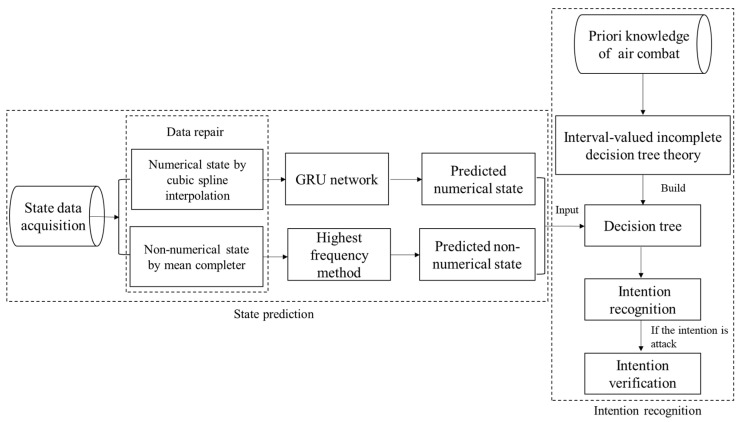
Intention recognition process.

**Figure 3 entropy-25-00671-f003:**
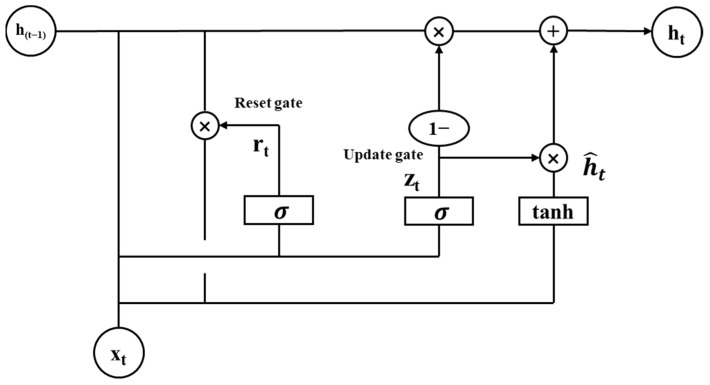
Structure of GRU network.

**Figure 4 entropy-25-00671-f004:**
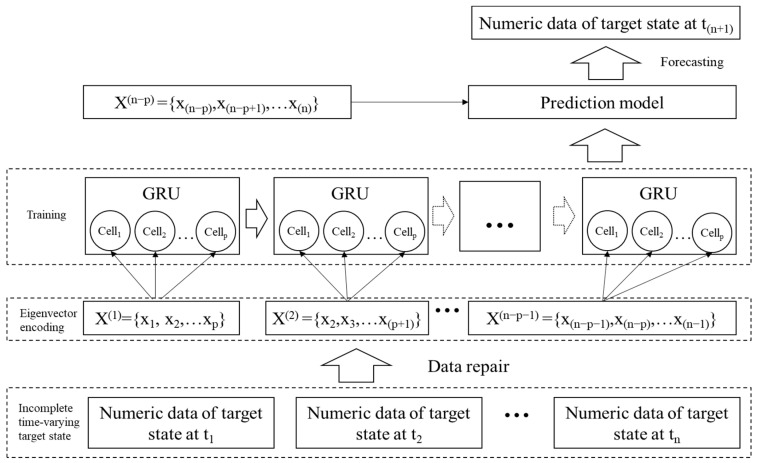
Framework of state prediction model.

**Figure 5 entropy-25-00671-f005:**
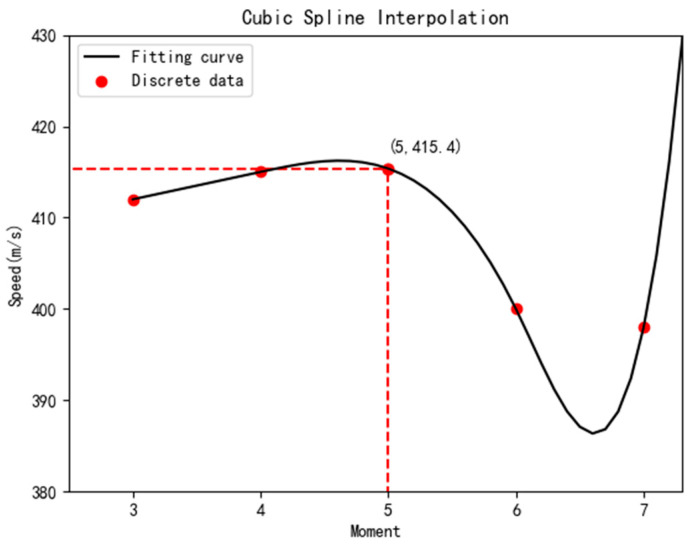
Data repair by cubic spline interpolation.

**Figure 6 entropy-25-00671-f006:**
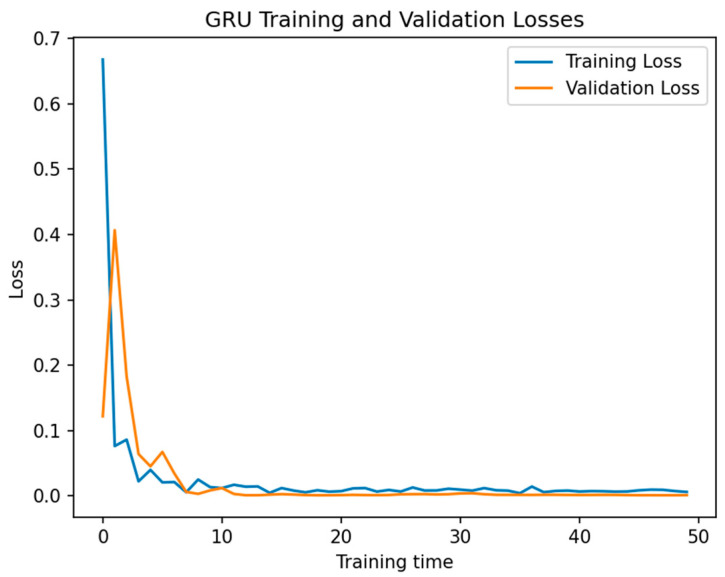
Training and validation losses.

**Figure 7 entropy-25-00671-f007:**
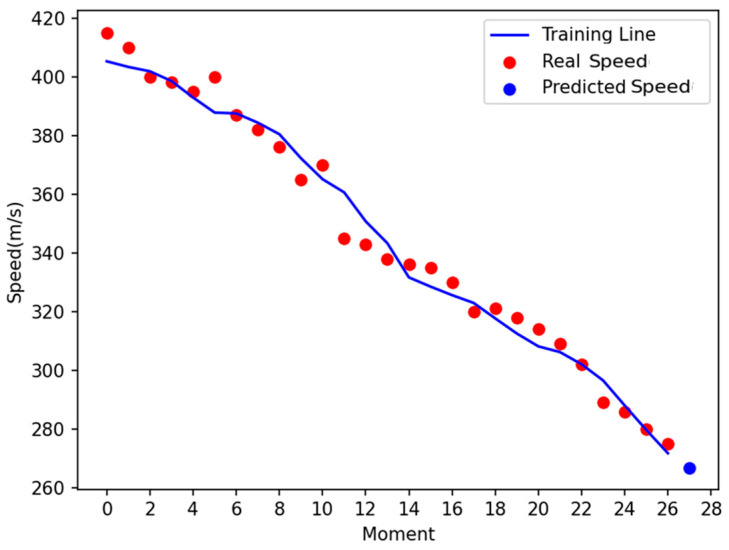
Fitting and prediction results.

**Figure 8 entropy-25-00671-f008:**
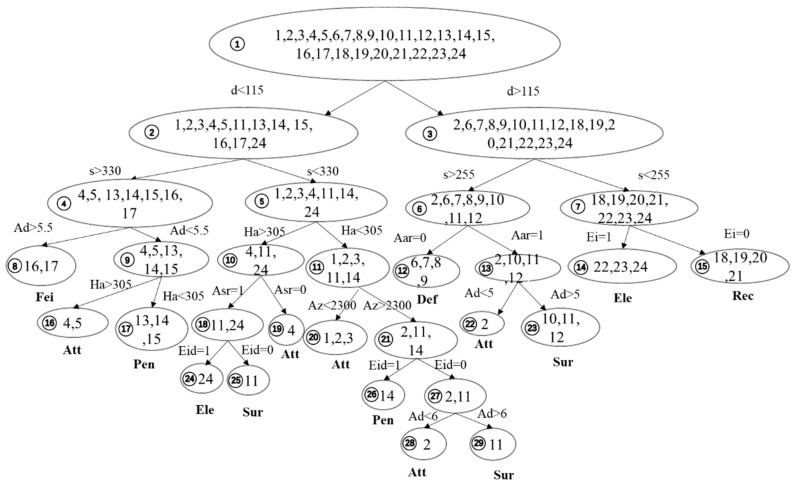
Intention decision tree.

**Table 1 entropy-25-00671-t001:** Numerical state data.

Moment	Speed (m/s)	Distance (km)	Altitude Difference (km)	Heading Angle (°)	Azimuth (mil)	Acceleration (m/s^2^)
1	420	395	14.6	35	755	4
2	415	392	14.3	34	750	2
3	412	386	14.4	30	*	1
4	415	382	14.6	*	765	−5
5	*	375	*	30	760	2
6	400	371	13.8	28	765	2
7	398	366	14	25	770	*
8	395	*	13.5	20	780	4
9	400	348	13.2	*	*	1
10	387	340	13	18	785	0
11	382	328	12.5	14.0	805	2
12	376	314	12.2	14.0	920	2
13	365	310	11	13.5	1020	-5
14	370	306	10.3	12.0	1005	3
15	345	301	10.5	13.0	1230	*
16	343	291	10.1	*	*	2
17	338	284	9.6	8.0	1400	3
18	336	267	*	10.0	1390	8
19	335	245	8.3	8.0	1380	5
20	*	230	7.6	7.0	1450	*
21	320	*	7.2	*	1560	5
22	321	195	6.3	9.0	1600	8
23	318	178	5.9	6.0	1750	10
24	314	169	5.2	5.0	1800	*
25	309	158	4.9	5.0	1850	-8
26	302	145	4.7	4.0	1930	10
27	289	132	4.2	1.0	*	-5
28	286	123	3.4	3.0	2050	9
29	280	115	2.6	1.0	2080	8
30	279	101	2.2	2.0	2110	8

Note: * indicates missing data.

**Table 2 entropy-25-00671-t002:** Non-numerical state data.

Moment	Air-to-Air Radar State	Air-to-Surface Radar State	Electromagnetic Interference State	Interfered State
1	1	1	0	0
2	1	1	0	0
3	0	1	0	0
4	1	*	1	1
5	0	0	1	1
6	*	1	1	1
7	1	1	*	0
8	1	1	1	1
9	1	*	*	1
10	*	0	1	1
11	1	1	*	*
12	1	1	0	0
13	1	1	*	0
14	1	0	0	0
15	*	1	1	0
16	1	1	1	0
17	1	1	1	*
18	1	0	0	1
19	0	1	1	0
20	1	1	1	1
21	1	1	1	1
22	0	1	0	*
23	1	1	1	1
24	1	0	1	1
25	*	1	*	1
26	1	0	1	0
27	1	*	1	*
28	*	1	1	1
29	1	1	1	0
30	1	1	1	1

Note: * indicates missing data, 0 indicates state “off”, 1 indicates state “on”.

**Table 3 entropy-25-00671-t003:** Test results of GRU network for speed.

Moment	25	26	27	28	29	30
Actual speed	309	302	289	286	280	275
Predicted speed	306.21	302.06	296.54	288.01	279.70	271.77
Relative error	−1.01%	+0.02%	+2.61%	+0.69%	−0.11%	−1.41%

**Table 4 entropy-25-00671-t004:** Network comparisons: GRU, LSTM, and RNN.

Computing Time (s)						
Model	Speed (m/s)	Distance (km)	Altitude Difference (km)	Heading Angle (°)	Azimuth (mil)	Acceleration (m/s^2^)
GRU	5.2	5.5	5.4	4.9	5.1	5.3
LSTM	7.1	8.1	6.7	6.8	7.5	7.8
RNN	7.8	7.9	6.7	7.2	7.3	7.3
**Root Mean Square Error**						
**Model**	**Speed (m/s)**	**Distance** **(km)**	**Altitude Difference** **(km)**	**Heading Angle (°)**	**Azimuth** **(mil)**	**Acceleration (m/s^2^)**
GRU	2.204	1.883	0.574	0.789	50.37	4.60
LSTM	3.068	2.898	0.571	0.879	62.53	4.72
RNN	7.868	5.367	0.927	1.2	103.25	6.65

**Table 5 entropy-25-00671-t005:** Numerical state data predicted by GRU network.

Moment	Speed (m/s)	Distance (km)	Altitude Difference (km)	Heading Angle (°)	Azimuth (mil)	Acceleration (m/s^2^)
31	266.87	92.31	2.06	1.43	2216.15	5.83

**Table 6 entropy-25-00671-t006:** Non-numerical state data predicted by HFM.

Moment	Air-to-Air Radar State	Air-to-Surface Radar State	Electromagnetic Interference State	Interfered State
31	1	1	1	1

**Table 7 entropy-25-00671-t007:** Intentions vs. numerical states.

No.	Speed (s)	Distance (d)	Altitude Difference (Ad)	Heading Angle (Ha)	Azimuth (Az)	Acceleration (a)	Intention (I)
1	[200, 260]	[50, 80]	[5, 6]	[20, 30]	[800, 1000]	[5, 10]	Att
2	[220, 270]	*	[2, 3]	[25, 40]	*	*	Att
3	[290, 320]	[70, 90]	*	[35, 45]	[2000, 2200]	[−5, 0]	Att
4	*	[40, 60]	[2.5, 3.5]	[330, 350]	[4000, 4100]	[7, 14]	Att
5	[340, 360]	[100, 110]	[3, 5]	[320, 330]	[4500, 4700]	[5, 8]	Att
6	[360, 380]	[300, 330]	[12, 13]	[100, 120]	*	[6, 10]	Def
7	[270, 300]	[270, 290]	[10, 11]	*	[2350, 2500]	[0, 3]	Def
8	[300, 320]	[270, 280]	[9, 10]	*	[5500, 5700]	[−10, −5]	Def
9	[360, 380]	[200, 220]	*	[210, 230]	[3900, 4150]	[5, 7]	Def
10	[355, 375]	[300, 330]	[7, 8]	*	[2300, 2500]	[−2, 2]	Sur
11	[280, 310]	*	[9, 10]	*	[3800, 4000]	[5, 8]	Sur
12	[370, 400]	[310, 350]	[10, 11]	[280, 300]	*	[0, 2]	Sur
13	[430, 450]	[80, 90]	[4, 5]	[100, 120]	[5300, 5400]	[12, 16]	Pen
14	*	[60, 70]	[2, 3]	[170, 190]	[2400, 2550]	*	Pen
15	[390, 410]	[50, 60]	[3, 4]	[280, 290]	[2350, 2500]	[15, 20]	Pen
16	[360, 370]	[90, 100]	[6, 7]	[30, 40]	*	[0, 2]	Fei
17	[350, 370]	[80, 90]	[7, 8]	[330, 350]	[4150, 4300]	[5, 9]	Fei
18	[150, 170]	[255, 270]	[6, 8]	[70, 80]	[5300, 5400]	[−1, 1]	Rec
19	[120, 140]	[200, 210]	[9, 10]	*	[1000, 1200]	[−8, −5]	Rec
20	[180, 190]	[150, 180]	*	[80, 90]	[2300, 2500]	[0, 2]	Rec
21	[220, 230]	[210, 220]	[7, 9]	[270, 290]	[4100, 4300]	[−10, −5]	Rec
22	[120, 140]	[180, 200]	[6, 7]	[20, 25]	[2300, 2400]	[0, 1]	Ele
23	[210, 230]	[120, 130]	[4, 5]	[30, 40]	[1050, 1200]	*	Ele
24	[230, 240]	*	[3, 4]	[320, 330]	*	[0, 2]	Ele

Note: * indicates missing data.

**Table 8 entropy-25-00671-t008:** Intentions vs. non-numerical states.

No.	Air-to-Air Radar State (Aar)	Air-to-Surface Radar State (Asr)	Electromagnetic Interference State (Ei)	Interfered State (Eid)	Intention (I)
1	1	0	1	0	Att
2	1	1	1	0	Att
3	1	*	1	1	Att
4	1	0	*	1	Att
5	*	1	*	0	Att
6	0	1	0	0	Def
7	0	1	1	*	Def
8	0	0	0	0	Def
9	0	1	1	0	Def
10	1	1	1	1	Sur
11	1	1	1	0	Sur
12	1	1	0	*	Sur
13	1	1	1	1	Pen
14	1	0	1	1	Pen
15	*	1	*	1	Pen
16	1	0	1	0	Fei
17	1	1	1	1	Fei
18	1	0	0	0	Rec
19	1	1	0	0	Rec
20	1	1	0	1	Rec
21	1	1	0	0	Rec
22	1	1	1	0	Ele
23	1	1	1	0	Ele
24	1	1	1	1	Ele

Note: * indicates missing data, 0 indicates state “off”, 1 indicates state “on”.

**Table 9 entropy-25-00671-t009:** Grading of numerical state vs. intention.

No.	Speed (s)	Distance (d)	Altitude Difference (Ad)	Heading Angle (Ha)	Azimuth (Az)	Acceleration (a)	Intention (I)
1	Medium	Short	Low	Small	East	Positive	Att
2	Medium	*	Low	Small	*	*	Att
3	Medium	Short	*	Small	East	Constant	Att
4	*	Short	Low	Small	West	Positive	Att
5	Fast	Medium	Low	Small	West	Positive	Att
6	Fast	Long	High	Large	*	Positive	Def
7	Medium	Long	Medium	*	South	Constant	Def
8	Medium	Long	Medium	*	North	Negative	Def
9	Fast	Medium	*	Large	West	Positive	Def
10	Fast	Long	Medium	*	South	Constant	Sur
11	Medium	*	Medium	*	West	Positive	Sur
12	Fast	Long	Medium	Medium	*	Constant	Sur
13	Fast	Short	Low	Large	North	Positive	Pen
14	*	Short	Low	Large	South	*	Pen
15	Fast	Short	Low	Medium	South	Positive	Pen
16	Fast	Short	Medium	Small	*	Constant	Fei
17	Fast	Short	Low	Small	West	Positive	Fei
18	Slow	Long	Medium	Medium	North	Constant	Rec
19	Slow	Medium	Medium	*	East	Negative	Rec
20	Slow	Medium	*	Medium	South	Constant	Rec
21	Medium	Medium	Medium	Medium	West	Negative	Rec
22	Slow	Medium	Medium	Small	South	Constant	Ele
23	Medium	Medium	Low	Small	East	*	Ele
24	Medium	*	Low	Small	*	Constant	Ele

Note: * indicates missing data.

**Table 10 entropy-25-00671-t010:** Split points and IEPs.

Split point	45	55	65	75	85	95	105	115	125
IEP	3.007	3.378	3.188	3.089	2.837	2.491	2.548	2.406	2.535
Split point	140	165	190	205	215	237.5	262.5	275	285
IEP	2.429	2.645	2.669	2.912	2.937	2.758	2.859	2.952	2.886
Split point	295	305	320	340					
IEP	2.797	2.906	3.205	3.007					

## Data Availability

Data is contained within the article. The data presented in this study are available in the article.

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
