# Peer review of "Air Combat Intention Recognition with Incomplete Information Based on Decision Tree and GRU Network"

_entropy, 2023, doi:10.3390/e25040671_

Round 1

Reviewer 1 Report

The paper covers interesting research on air combat intention recognition with incomplete information based on decision trees and the GRU network. The manuscript does not analyze all possible battle scenarios. The most rational alternatives can be lost without developing and analyzing all possible battle scenarios. However, the manuscript is written in style more like a report rather than a research article. The global innovativeness in research development hasn't been presented. Some figures and tables which involve world-wide novel research should be described and discussed with more details to emphasize the state-of-the-art-review all over the world novelty. Please use this the newest (2019-2023) Web of Science journal papers.

Author Response

Please see the attached PDF document.

Reviewer 2 Report

The paper is written in a good style, good English. 

The results are substantiated, supported with experiments.

The only comment is that the majority of the bibliorgaphy in the lit. review refer to Chinese authors, however, the contribution of the other authors in these fields are sometimes much more important.

Author Response

Please see the attached PDF document.

Reviewer 3 Report

The authors submitted an exciting article that proposed a model based on a decision tree for recognizing the intentions of enemy fighters in combat.
Developing support weapon systems is highly relevant in today's increasing war tensions and geopolitical interests.
Decision-making tools based on artificial intelligence can significantly help design a combat strategy.
I have a few notes for the authors:
1. Authors should format the article according to the MDPI template.
MS Word: https://www.mdpi.com/files/word-templates/entropy-template.dot
LaTeX: https://www.mdpi.com/authors/latex
2. What is the shape of the resulting cubic spline model for data integration?
3. The authors' calculation and simulation software is unclear.
4. What are the possibilities of adaptation of the prediction model? Please discuss it more.
5. What was the prediction model's performance on the training data, and what on the test (unknown) data? Please discuss this more and illustrate the differences.
6. How will the performance of the predictive model change when the number of input variables is changed and the training set be increased?
7. Authors should discuss the results and how they can be interpreted from the perspective of previous studies and the working hypotheses. The findings and their implications should be discussed in the broadest context. Future research directions may also be highlighted. The practical use of the proposal should be emphasized more.
I suggest accepting the paper after revision.
Rewriting to the MDPI template is mandatory.
I wish the authors success.

Author Response

Please see the attached PDF document.
